# Identification and characterization of calcium binding protein, spermatid-associated 1 (CABS1)[#] in selected human tissues and fluids

Eduardo Reyes-Serratos[1,2], Joy Ramielle L. Santos[1,2], Lakshmi Puttagunta[1,3], Stephen J. Lewis[4], Mechiko Watanabe[4,5], Aron Gonshor[6], Robert Buck[6,7], A. Dean Befus[1,2]*, Marcelo Marcet-Palacios[1,8]*

1 Alberta Respiratory Centre, University of Alberta, Edmonton, Alberta, Canada, 2 Division of Pulmonary Medicine, Department of Medicine, University of Alberta, Edmonton, Alberta, Canada, 3 Department of Laboratory Medicine, University of Alberta, Edmonton, Alberta, Canada, 4 Departments of Pediatrics and Pharmacology, Rainbow Babies and Children's Hospital, Case Western Reserve University, School of Medicine, Cleveland, Ohio, United States of America, 5 Division of Pediatric Cardiology, Department of Pediatrics, Rainbow Babies and Children's Hospital, Case Western Reserve University, School of Medicine, Cleveland, Ohio, United States of America, 6 GB Diagnostics, Montreal, Quebec, Canada, 7 GB Diagnostics, Albuquerque, New Mexico, United States of America, 8 Northern Alberta Institute of Technology, Edmonton, Alberta, Canada

* marcelo@ualberta.ca (MMP); dean.befus@ualberta.ca (ADB)

**Data Availability Statement:** All relevant data are within the manuscript and its Supporting information files. Also the public repository can be

## Abstract

Calcium binding protein, spermatid associated 1 (CABS1) is a protein most widely studied in spermatogenesis. However, mRNA for CABS1 has been found in numerous tissues, albeit with little information about the protein. Previously, we identified CABS1 mRNA and protein in human salivary glands and provided evidence that in humans CABS1 contains a hepta-peptide near its carboxyl terminus that has anti-inflammatory activities. Moreover, levels of an immunoreactive form of CABS1 were elevated in psychological stress. To more fully characterize human CABS1 we developed additional polyclonal and monoclonal antibodies to different sections of the protein and used these antibodies to characterize CABS1 in an overexpression cell lysate, human salivary glands, saliva, serum and testes using western blot, immunohistochemistry and bioinformatics approaches exploiting the Gene Expression Omnibus (GEO) database. CABS1 appears to have multiple molecular weight forms, consistent with its recognition as a structurally disordered protein, a protein with structural plasticity. Interestingly, in human testes, its cellular distribution differs from that in rodents and pigs, and includes Leydig cells, primary spermatogonia, Sertoli cells and developing spermatocytes and spermatids, Geodata suggests that CABS1 is much more widely distributed than previously recognized, including in the urogenital, gastrointestinal and respiratory tracts, as well as in the nervous system, immune system and other tissues. Much remains to be learned about this intriguing protein.

found at: https://sites.ualberta.ca/~marcelo/Data_Repository.zip.

**Funding:** MMP, RGPIN-2020-04553, Natural Sciences and Engineering Research Council of Canada, https://www.nserc-crsng.gc.ca/. The funder had no role in study design, data collection and analysis, decision to publish, or preparation of the manuscript. ADB, 16BB-MSI-C5 and 18BB-SI-C9. AllerGen. https://www.allergen.ca/. The funder had no role in study design, data collection and analysis, decision to publish, or preparation of the manuscript. GB Diagnostics, 10515302 Canada Inc. provided funding within the company for characterization of the polyclonal and monoclonal antibodies to human CABS1 and for their long term storage. Drs Buck and Gonshor, co-owners of GB Diagnostics contributed to design of the monoclonal antibodies and analysis of their specificities, as well as to discussions about the preparation of the manuscript and its editing. Stephen Lewis and Michiko Watanabe have no funding to declare.

**Competing interests:** I have read the journal's policy and the authors of this manuscript have the following competing interests: A. Dean Befus, together with the University of Alberta holds a patent on CABS1 as a biomarker of stress, and an associated licensing agreement with GB Diagnostics, 10515302 Canada Inc. Aron Gonshor and Robert Buck are co-owners of GB Diagnostics 10515302 Canada Inc. GB Diagnostics helped with the processing of the application and partially funded the approved USA patent 16/084,617, entitled: Calcium Binding Protein, Spermatid Specific 1, as a Biomarker for Diagnosis or Treatment of Stress. The company is also assisting with the application and partially funding of the Canadian National Application No. CA 3,017,604, Calcium Binding Protein, Spermatid Specific 1, as a Biomarker for Diagnosis or Treatment of Stress (pending approval). GB Diagnostics 10515302 Canada Inc. is developing a stress biomarker test for commercial application under its licensing agreement with the University of Alberta. The commercial affiliation with GB Diagnostics Canada Inc. does not alter our adherence to PLOS ONE policies on sharing data and materials for this manuscript.

## Introduction

Using an experimental model in rats to investigate the neural control of inflammatory reactions, we identified a superior sympathetic-submandibular gland axis with anti-inflammatory activity [1–4]. Ultimately, we established that the protein SMR1 (Submandibular rat 1) contains a seven amino acid peptide (TDIFEGG) near its carboxyl terminus with anti-inflammatory activity [3]. Previous work had determined that SMR1 expression was sexually dimorphic, much more abundant in male rats than females, and present in the submandibular glands and testes [5]. SMR1 was predicted to be a prohormone with several polypeptide derivatives with multiple functions [6–11], including vasculature/erectile function, analgesia, and mineralocorticoid-like activity. Our identification of the anti-inflammatory activity was novel, and we characterized it in several models, including immediate hypersensitivity [12], endotoxic shock [13], asthma [14, 15], and spinal cord injury [16].

However, an analysis of the human genome showed that SMR1 was absent [17], and thus to determine if the anti-inflammatory axis identified in the rat existed in humans, we tested if any human protein contained a heptamer with a similar sequence to that of TDIFEGG in the rat [4, 17, 18]. Human calcium-binding protein, spermatid-associated 1 (CABS1), was identified given a heptamer TDIFELL near its carboxyl terminus, which was also shown to have anti-inflammatory activity [18]. Moreover, as with SMR1, CABS1 was expressed in testes [19–21], and we identified it in submandibular glands as well [18]. CABS1 was found on the human chromosome analogous to rat chromosome 14, namely chromosome 4, in a region sharing several analogous genes to those in the rat [17].

We collaborated with an experimental psychologist, Thomas Ritz, to determine if human CABS1 was under autonomic control like SMR1. We established that levels of immunogens in saliva that reacted with anti-CABS1 polyclonal antibody (pAb) were associated with acute experimental stress [22]. To extend these studies, we developed additional pAb and monoclonal antibodies (mAb) to human CABS1 (hCABS1). The current study characterizes these antibodies using extracts of testes and submandibular glands, and saliva, serum and human CABS1 overexpression lysate from cultured cells. Our results were then validated using immunohistochemistry to localize CABS1 in selected human tissues, and bioinformatics analyses of CABS1 mRNA expression from public data (GEOdata). Some of this information has been reported in the PhD thesis of Eduardo Reyes-Serratos [23].

## Materials and methods

### Antibodies to human CABS1

Polyclonal and monoclonal antibodies were produced in accordance with international standards. The University of Alberta guidelines, in accordance with the Canadian Council on Animal Care, do not mandate an ethics review of antibody production by accredited commercial suppliers. Our supplier, GenScript Biotech, Piscataway, NJ, USA, is fully accredited by the Association for Assessment and Accreditation of Laboratory Animal Care and NIH Office of Laboratory Animal Welfare.

**Polyclonal antibodies.** Rabbit pAb were made against two regions of CABS1 (Fig 1). Polyclonal antibody (pAb) H1 was raised against hCABS1 amino acid (aa) 375–388 TSTTETDIFELLKE (underlined anti-inflammatory sequence [18]). Polyclonal antibodies H2.0, H2.1, and H2.2 were raised against hCABS1 aa 184–197 DEADMSNYNSSIKS (region deemed to be highly immunogenic for B lymphocytes).

**Monoclonal antibodies.** Mice were immunized against peptide sequences of hCABS1 (Fig 1). Their spleens were harvested, and spleen-derived B-cells were combined with

## Immunogens used to induce antibodies to CABS1

**Fig 1. The ranges of the amino acid sequence immunogens are indicated across the top row and the mAb and pAb are listed in the bottom rows.** C = carboxyl terminal, N = amino terminal.

myelomas to create hybridomas (GenScript Biotech). Monoclonal antibody 13G3 was raised against hCABS1 aa 375–388 TSTTETDIFELLKE (underlined anti-inflammatory sequence). Monoclonal antibody 15B11 was raised against CABS1 aa 184–197 DEADMSNYNSSIKS. The immunogen for monoclonal antibody 4D1 was the complete recombinant hCABS1 protein. The mAb detects a region towards the N terminus (aa 40–60 S1 Fig).

### CABS1 transient overexpression cell lysate

A recombinant hCABS1 overexpression lysate (OEL) produced in Human Embryonic Kidney 293T (HEK293T) cells (OriGene Technologies Inc., Rockville, MD, USA) was used as a positive control in Western Blot (WB). The lysate contains recombinant hCABS1 protein with a FLAG tag (aa sequence: DYKDDDDK) adjacent to its carboxyl end. The negative control cell lysate (NCL) was from HEK293T cells with the same vector but lacking a hCABS1 cDNA insert. As an additional control, we probed the overexpression lysate with ANTI-FLAG [®] M2 (Sigma), a mouse mAb targeting FLAG sequence (gift from Drs. Steven Willows and Marianna Kulka, Nanotechnology Research Centre, Edmonton, AB).

### Human samples

Unless otherwise stated, all tissues and fluids were collected and stored under ethics protocols approved by the University of Alberta Human Ethics Review Board, (Biomedical Panel). The Alberta Research Information Services (ARISE) System approved our protocols Pro00001790 (Anti-inflammatory proteins and biomarkers of stress) and Pro00112432 (Distribution and cell-type specific localization of CABS1) in writing. Protocol Pro00001790 was initially approved in 2007 and has been subject to annual review and re-approval as well as amendments. Protocol Pro00112432 was approved in 2021 and was annually reviewed and renewed in 2022 and 2023. As the human submandibular gland (SMG) and testes samples used in this study were archived surgical specimens collected for clinical diagnostic purposes, our protocol was for secondary use of the samples and received a waiver of consent from the Ethics Review Board.

Four samples of human SMG were purchased from the commercial source, Conversant Biologic Inc (see below). S2 Table lists all the testes and submandibular gland samples used in this study.

### Saliva and serum

Recruitment for a pool of saliva and serum, each from the same individual, a 30-year-old male, started January 1, 2017, and completed June 30, 2018. Written consent was received from the study subject.

Whole blood (10 mL) was collected in a vacutainer containing no additive, incubated for 45 min to allow clotting and centrifuged at 1500 g, 21°C for 15 min. Serum was aspirated, transferred into 300 μL aliquots, and stored at -80°C.

Unstimulated whole saliva was collected using the passive drool technique protocol (Salimetrics LLC, Carlsbad, CA, USA) [24] over 15 days. Following breakfast and teeth brushing, saliva was collected after 1.5 h. Samples were frozen at -20°C immediately after collection. When the collection was finalized (200 mL), all samples were thawed and pooled. The resulting pool was centrifuged at 1500 g, 4°C for 20 min, and the supernatant was transferred into 50 μL aliquots. Prior to analyses, total protein concentration was determined as below.

## Submandibular gland (SMG)

For tissue lysates to be used for WB, fresh human SMG samples were collected between January 1, 2009 and June 30, 2018 (n = 8, S2 Table) from patients, female and male, undergoing surgical removal of squamous carcinoma at the University of Alberta Hospital (Edmonton, Canada). A grossly normal looking portion of each SMG was homogenized at 4°C using a diluent RIPA buffer supplemented with P8340 protease inhibitor cocktail (Sigma Aldrich, Markham, ON, Canada). The remaining fresh tissue was frozen (-80°C) for future lysate extraction. Post-homogenization, samples were centrifuged at 12,000 g for 20 min, and the supernatant was collected, aliquoted, and stored at -80°C. Before WB analyses, each sample's total protein concentration was determined using a Pierce™ BCA Protein Assay Kit (Thermo Scientific, Waltham, MA, USA).

For immunohistochemical studies surgical specimens of SMG were collected through the University of Alberta (n = 3) (secondary use, Pro00112432). The specimens were identified between July 30, 2021 to November 17, 2023 from archived samples that were originally collected for clinical diagnostic purposes. Two samples were from females, aged 62 and 73, and one male, aged 69 (S2 Table). SMG samples were fixed in 10% buffered formalin. Following paraffin embedding, tissue sections were assessed by microscopy using hematoxylin and eosin (H&E) staining to identify areas that appeared to be morphologically normal. Thereafter, adjacent sections were stained with anti-hCABS1 antibodies and normal areas were assessed for the distribution of CABS1 immunoreactivity.

Four specimens of submandibular gland, two from carcinoma resection (one female 40 years and one male 62 years), and two from Sjogren's syndrome subjects, a 43-year-old male and an unknown donor (S2 Table), were kindly provided to Dr. Stephen Lewis by GlaxoSmithKline's (GSK) Immune-inflammation therapy unit's human biological sample repository, arranged by Dr. Alessandra Giarola of Bioelectronics Research and Development. The samples were purchased from Conversant Biologics Inc, now Discovery Life Sciences, Huntsville, AL, USA 35806.

## Testes

Archived surgical specimens (n = 4) of testes tissue from 58, 70 and 73 year old, and one of unknown age were used for immunohistochemical studies (S2 Table).

## Western blot analysis

For 1 D electrophoresis, we followed standard BIO-RAD protocols. Briefly, the TGX™ FastCast™ 12% Acrylamide Kit (Bio-Rad, Mississauga, ON, CA) was used to prepare 1.5 mm thick gels the day before an experiment and polymerized gels were kept at 4°C. On the day of the experiment, 1X WB running buffer (Trizma base– 40 mM, glycine– 300 mM, methanol– 200 mL) was prepared and kept at 4°C.

For WB, electrophoresed samples in gels were placed in a cassette containing fiber pads, filter papers, and a 0.45 μm pore-size nitrocellulose membrane. The cassette was placed in a vertical transfer chamber with an ice pack on the side in WB transfer buffer; transfer was done at 0.5 A for 1 h. The nitrocellulose membrane was washed with double distilled water before adding WB blocking buffer (1% fish skin gelatin [Truin Science Ltd., Edmonton, AB, Canada], 1X PBS, 0.1% Tween20). Blocking was done at room temperature for 1 h. The membrane was probed overnight at 4°C with the relevant titrated antibody(-ies) diluted in a 50:50 solution containing PBS supplemented with 0.05% Tween 20, and blocking buffer. The antibody solution was then removed, and the membrane was washed. LI-COR Biosciences secondary antibodies were used at 0.05 μg/mL in the same solution as the primary antibodies. The dates when the WB experiments were conducted, together with the data files are included in the link data repository folder, WB section, https://sites.ualberta.ca/~marcelo/Data_Repository.zip.

The controls for the pAbs included: no primary antibody just secondary goat anti-mouse antibodies (LI-COR, Lincoln, NE, USA) and for some tissues, preimmune serum; for mAbs included: no primary antibody, or isotype antibodies concentration-matched to our mAbs. Mouse IgG1.κ (R&D systems, Minneapolis, MN, USA) was the isotype antibody for mAbs 15B11 and 13G3; mouse IgG2α.κ (Stemcell™ Technologies, Vancouver, BC, Canada) was the isotype antibody for mAb 4D1. There is a repository of detailed blot/gel data together with a WB protocol in the linked data repository folder, WB section.

## Molecular weight determination

All WB included a molecular mass reference ($M_r$). WB images from studies of pAb produced by LI-COR were opened in Microsoft Paint and the $M_r$ of each band (kDa) was recorded. For $M_r$ estimates from studies with mAb, calculations were done manually.

## Immunohistochemical analyses (immunoperoxidase or immunofluorescence amplification)

**SMG and testis.** For human tissues collected at U of A from archives, immunohistochemical studies used immunoperoxidase amplification. Sections on slides were baked at 60°C for 1 h and then deparaffinized and retrieved on Dako Omnis fully automated IHC platform (Agilent, Santa Clara, CA, USA) using EnVision FLEX target retrieval solution, high pH (Dako Omnis) at 97°C for 30 min. Monoclonal antibody 15B11 (targeting the mid region of hCABS1) was diluted 1/100 in EnVision FLEX antibody diluent and was applied for 20 min. at 32°C followed by EnVision Flex+ Mouse LINKER (Dako Omnis) for 10 min. Negative controls included the use of no anti-CABS1 mAb, or isotype control antibody as described above for WB studies. Subsequently, EnVision FLEX, HRP.Rabbit/Mouse was added for 20 min. Visualization was performed using DAB+ substrate chromogen system (Dako Omnis) and counter-stained with Gill I Hematoxylin formula (Leica, Wetzlar, Hesse, Germany). All histology and immunostains were reviewed with an Olympus BX41. All histology images were captured with a Nikon Eclipse E600 microscope with an attached Nikon Digital-sight DS-Fi2 Ki7607 camera.

Following optimization of the staining protocols on testis tissue using three dilutions of 15B11 anti-CABS1 mAb, the immunohistochemical studies of SMG and testes used three slides of tissue sections for each subject. One slide was used for the test mAb and two slides for the negative controls; no primary mAb or concentration-matched isotype control. The average area of tissue assessed was: 4.9 cm$^2$ for SMG and 4.0 cm$^2$ for testes. Three co-authors (LK, ADB and MM-P) reviewed the slides and reached consensus about the results.

For immunofluorescence studies, SMG tissue (S2 Table) (gift from GSK, see above) was sectioned, deparaffinized, rehydrated, and microwaved for 10 min in 10 mM sodium citrate (pH 6) for antigen retrieval (i.e., heat-induced antigen retrieval). Subsequently, sections were blocked for 30 min at room temperature in blocking buffer (PBS containing 5% bovine serum albumin and 0.1% Triton-X 100). Next, blocked sections were incubated overnight at 4˚C with the primary pAb (H1.0, H2.0, H2.1, H2.2, or respective preimmune serum, diluted 1/200 in blocking buffer. The next day, slides were washed with PBS and incubated for 2 h at room temperature with the secondary antibody (goat anti-rabbit IgG H+L conjugated with Alexa Fluor 488, Thermo Scientific), washed with PBS and mounted in VECTASHIELD® HardSet™ anti-fade medium with DAPI (Vector Laboratories, Burlingame, CA, USA). Images were acquired with a Retiga EXi, Fast 1394 digital camera (Teledyne Photometrics, Tucson, AZ, USA) attached to a Diaphot 200 inverted phase contrast microscope (Nikon) and evaluated using Q-Capture imaging software (Teledyne Photometrics).

For studies with anti-CABS1 pAb H1.0, H.2.0, H2.1 and H2.2, one slide from each specimen for each pAb was assessed. For each preimmune serum negative control, one slide from each specimen was assessed. Four observers reviewed the slides (two co-authors MW and SL, one trainee and one technologist) and reached consensus about the results. S3.1–S3.8 Fig provide additional images of the immunofluorescence studies of submandibular glands.

## Mass Spectrometry sequencing (MS-seq)

Transient overexpression samples, OEL and NCL, were separated by 1-D SDS-PAGE and stained using Blue Silver dye. To assess the distribution and relative abundance of hCABS1, the resolving gels (complete lanes) were cut into 11 equal segments and sent for analysis to the Alberta Proteomics and Mass Spectrometry Facility (University of Alberta, Edmonton) for in-gel trypsin digestion and MS-seq analysis using a LTQ Orbitrap XL Hybrid Ion Trap-Orbitrap mass spectrometer (Thermo Fisher Scientific). Data were processed using Proteome Discoverer v.1.4 (Thermo Fisher Scientific) using the Sequest (Thermo Fisher Scientific) database search algorithm; only proteins with $\geq 2$ tryptic peptides identified by MS-seq were included. Data on all the peptide fragments detected during our MS-seq studies are attached in the linked data repository folder, mass spectroscopy section, https://sites.ualberta.ca/~marcelo/Data_Repository.zip.

## Meta-analysis of studies from Gene Expression Omnibus (GEO) database

Gene expression data was collected from the GEO database and supplemental data provided in previous studies. Relevant studies were identified using specific keywords, including CABS1, Normal, and Tissue. Non-normal tissue or treated cell/tissue datasets were excluded from the search. Raw data was downloaded and rescaled to 0 to 100% for each dataset using min-max normalization. We combined the data from three studies [25–27] and analyzed the data distribution in a histogram. Over 94% of the data points were observed within 0 to 50%. A continuous graded color scale of the data distribution was used to create the meta-analysis figure.

## Results

### Detection of recombinant hCABS1 in OEL using pAb and mAb

**pAb.** We previously published studies of hCABS1 in OEL and NCL using pAb H2.0 [18]. These studies were confirmed and extended with pAb H1.0 and H2.0, H2.1, and H2.2 (Fig 2A, reported in PhD thesis, [23]). In addition to the NCL control, the results with the anti-FLAG antibody and the no primary antibody are also shown (Fig 2A). H1.0, H2.0, and H2.2

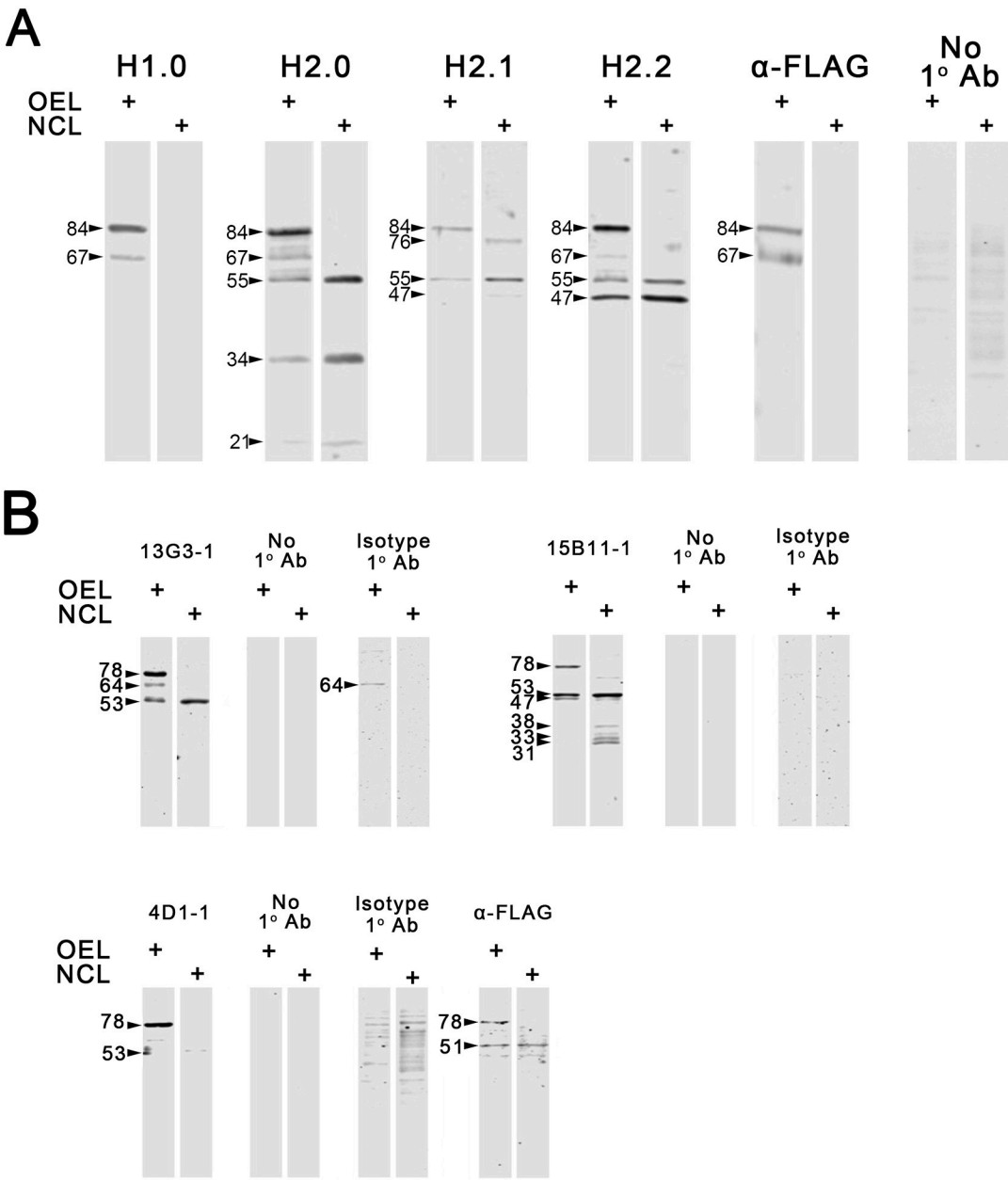

**Fig 2. Western blot analyses of human CABS1 in transient overexpression lysate (OEL) of HEK 293T cells and in control lysate (NCL) of HEK 293T cells transfected with empty plasmid.** A) Representative results of four pAb to peptide immunogens of hCABS1 are shown (n = 8 to 12 replicates; 1 μg of OEL or NCL lysate were used, for H1.0 and H2.0, 3 ng/μL and for H2.1 and H2.2, 2 ng/μL of pAb were used). Negative control studies using no primary antibody were clean. In B) mAb 13G3 (10 ng/μL), 15B11 (1 ng/μL) and 4D1/ (1ng/μL) were used, and 1 μg of OEL or NCL (n = 3). Controls with no primary antibody, or with isotype control antibodies are shown. Antibody to the FLAG tag on the rhCABS1 was used as a control for these studies with pAb and mAb.

specifically detected immunoreactive bands in OEL at 84 and 67 kDa, whereas H2.1 detected the 84 kDa band specifically (Fig 2A; n = 8–12 independent experiments). In NCL, these pAbs variably detected bands at approximately 76, 55, 47, 34 and 21 kDa (Fig 2A), but nothing at 84 and 67 kDa. Antibody to the FLAG marker of rhCABS1 in OEL detected immunoreactive

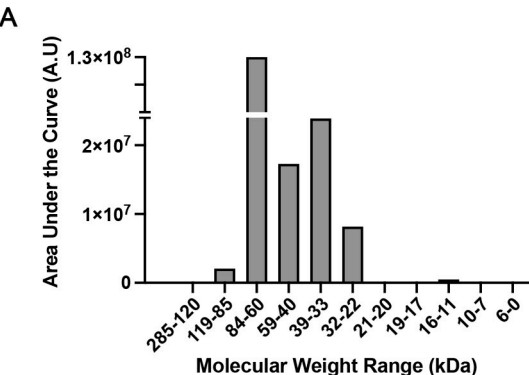

**Fig 3. Estimate of the abundance of rhCABS1 in sections (range of kDa) of 1 D gel analyzed by mass spectroscopy.**
A) relative abundance of rhCABS1 in each gel segment (kDa range); B) the number of rhCABS1 peptides detected in
each gel segment, the percentage of rhCABS1 covered by the peptides in each segment and the percentage of the total
rhCABS1 recovered. In summary, our pAbs detected CABS1-specific bands at 84 and 67 kDa in rhCABS1 OEL and
not in NCL.

bands at 84 and 67 kDa; no bands were detected in NCL. Similarly, MS-seq analyses detected
rhCABS1 in OEL, but not in NCL. CABS1 peptides were detected in 1D gel segments of 119–
85, 84–60, 59–40, 39–33, 32–22 and 16–11 kDa (Fig 3), as mentioned briefly in our previous
report [18]. Two to 10 unique CABS1 peptides were identified in the gel segments (Fig 3B) and
peptide coverage of the protein in the segments ranged from 7 to 48%. As previously reported
[18], there was no evidence that peptides from selected regions of hCABS1 were preferentially
found in different segments of the gel. No peptides were detected above 119 kDa, in the 17–21
range or below 10 kDa. The linked data repository folder, mass spectroscopy section has the
raw MS-seq data of the peptides from various proteins that were detected in OEL and NCL.

**mAb: (n = 3 independent experiments for each).** Antibody 13G3 (same immunogen as
H1 pAb) detected bands at 78, 64, and 53 kDa in OEL, and in NCL, a band at 53 kDa (Fig 2B).
No bands were observed in membranes probed with no primary antibody, but OEL probed
with isotype antibody IgG1.κ at the same concentration as 13G3 showed a band at 64 kDa (Fig
2B). WB analyses using 15B11 (same immunogen as H2.n pAb) showed immunoreactive
bands at 78, 53, and 47 kDa for OEL, and bands at 53, 47, 38, 33, and 31 kDa in NCL (Fig 2B).
No immunoreactivity was observed in membranes probed with isotype antibody IgG1.κ at the
same working concentration as 15B11 or with no primary antibody. Antibody 4D1 shows a
band at 78 kDa in OEL, and a faint band at 53 kDa in NCL (Fig 2B). No signal was detected in
membranes probed with no primary antibody, but a membrane probed with isotype antibody

IgG2α.κ showed several bands for NCL in a smear pattern in a range between 100 and 35 kDa. Evaluation of these lysates with anti-FLAG antibody showed two bands at 78 and 51 kDa, with the 51 kDa band also present in NCL (Fig 2B).

In summary, our mAbs specifically detected rhCABS1 at 78 kDa in OEL. Given that this band was also FLAG positive, we postulate that the 84 kDa band detected with pAbs and this band detected at 78 kDa with mAbs are the same (see Discussion).

## Detection of CABS1 using pAbs and mAbs in human fluids and submandibular glands lysate

**hCABS1 in submandibular gland lysate.**   *pAb*. With each of the four pAb several immunoreactive bands ranging from ~22 to 127 kDa were detected in male SMG lysate (Fig 4A). Controls using no primary antibody were negative. There were differences among the pAb in the immunoreactive bands that were detected; bands at ~71, 52 and 27 kDa were detected by at least three of four pAb.

*mAb*. No immunoreactive bands were detected when probing membranes with isotype antibody at the same working concentration as the respective primary mAb, or when adding no primary antibody. WB analyses of male and female SMG lysates using 13G3 (n = 3) showed immunoreactive bands at ~96, 78, 63, 53 and 42 kDa. (Fig 4B). WB analyses of SMG lysate using the 15B11 mAb (n = 3 independent experiments) detected a single immunoreactive band at 53 kDa in both male and female extracts. WB of SMG lysate using antibody 4D1 (n = 3) detected two bands at 63 and 53 kDa in both male and female SMG, and in the male, additional bands at 96 and 78 kDa and between 63 and 53 kDa (Fig 4B).

Thus, with mAbs, SMG lysate has anti-hCABS1 specific immunoreactive bands at ~96, 78, 63, 53 and 42 kDa with some distinctions among the mAbs used. With pAb, bands at ~71, 52 and 27 kDa were detected with 3 of 4 pAb.

## hCABS1 in saliva

**pAbs.**   Given our experience with careful antigen and antibody titrations using multiple human saliva samples with WB [18], and as well many of the same saliva samples with semi-automated nano-capillary immunoassay (NCIA) [24], we reassessed our previous optimization [18] of antibody and antigen concentrations using pAb and the pooled human saliva from a single individual. We determined that optimal specificity was achieved at 2.2 μg of human saliva and 0.2 to 0.3 ng/μL of pAbs. At these concentrations in a pilot experiment, no unequivocal bands selective to H1 compared to preimmune serum or no primary antibody controls were detected in saliva, although a band was seen at 60 kDa (Fig 5A). With H2.0 a selective band (versus preimmune serum) was seen at approximately 60 kDa and faint bands were seen at ~ 90 and 27 kDa (Fig 5A). Studies were not done with H2.1 and 2.2.

**mAbs.**   As with studies using pAb, studies with all three mAb on salivary supernatants detected a number of immunoreactive bands. However, many of them appear to be non-specific and the only specific immunoreactive band was at ~ 60 kDa; not detected with only secondary antibody or with isotype antibody (Fig 5B).

In summary, results from WB analyses of human saliva with pAb and mAb suggest that there is a specific immunoreactive band of ~60 kDa (see Discussion).

## hCABS1 in serum

**pAbs.**   For analysis of CABS1 in the pool of human serum and given our experience with saliva and in initial pilot experiments with pAbs, we selected 1.5 μg of serum and 0.03 and 0.02 ng/μL of pAbs H.1.0, H2.0, and H2.1, H2.2 respectively as appropriate concentrations of

# Submandibular Gland

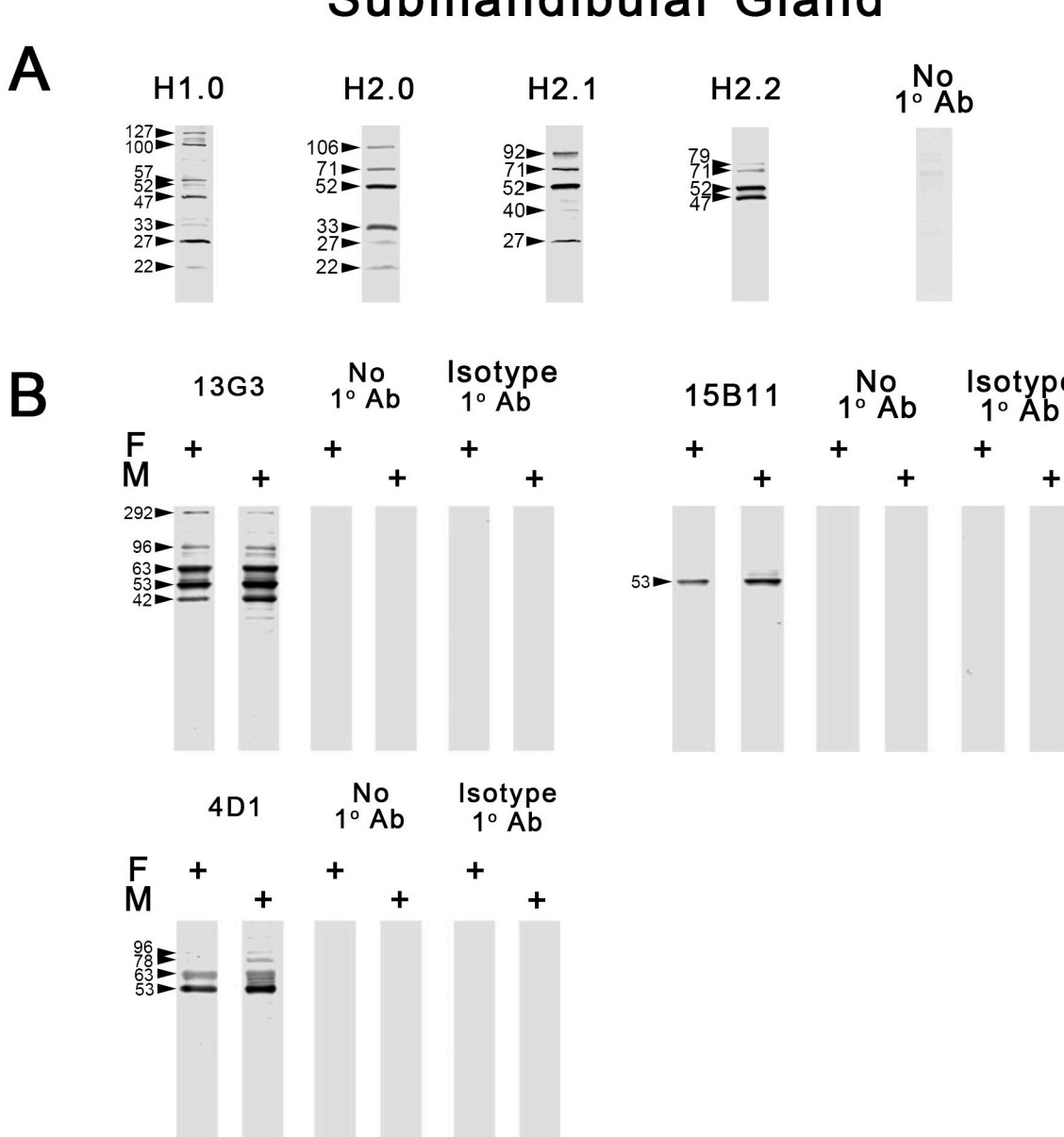

**Fig 4. Western blot analyses of CABS1 in human female and male submandibular gland lysates.** A) As optimized during studies with OEL and NCL (Fig 2), H1.0 and H2.0 (3 ng/μL) and H2.1 and 2.2 (2 ng/μL) were used for studies of male SMG lysate (25 μg) (n = 6–9). Controls included no primary antibody, n = 3. B) Studies with mAb used 10 μg of lysate in each lane; for mAb 13G3 and 4D1 10 ng/μL and for 15B11, 1 ng/μL of mAb was used (n = 3). M is male and F is female. Controls with no primary mAb or with isotype control mAb are shown and were negative.

reagents (S2 Fig). Several immunoreactive bands were detected with the pAbs that were not detected in the corresponding preimmune sera (S2 Fig). The bands ranged from ~ 180–135, 90–100, 60–75 and 47–56 kDa. Given this complexity of immunoreactive bands with pAb, we focused studies on mAb. Controls with no primary antibodies were clean.

**mAbs.** For studies of CABS1 immunoreactivity in the serum pool, 1 μg of serum was used and 10 ng/μL of mAb (n = 3 for each mAb). When probing serum with 13G3 and 4D1, WB

# Human Saliva

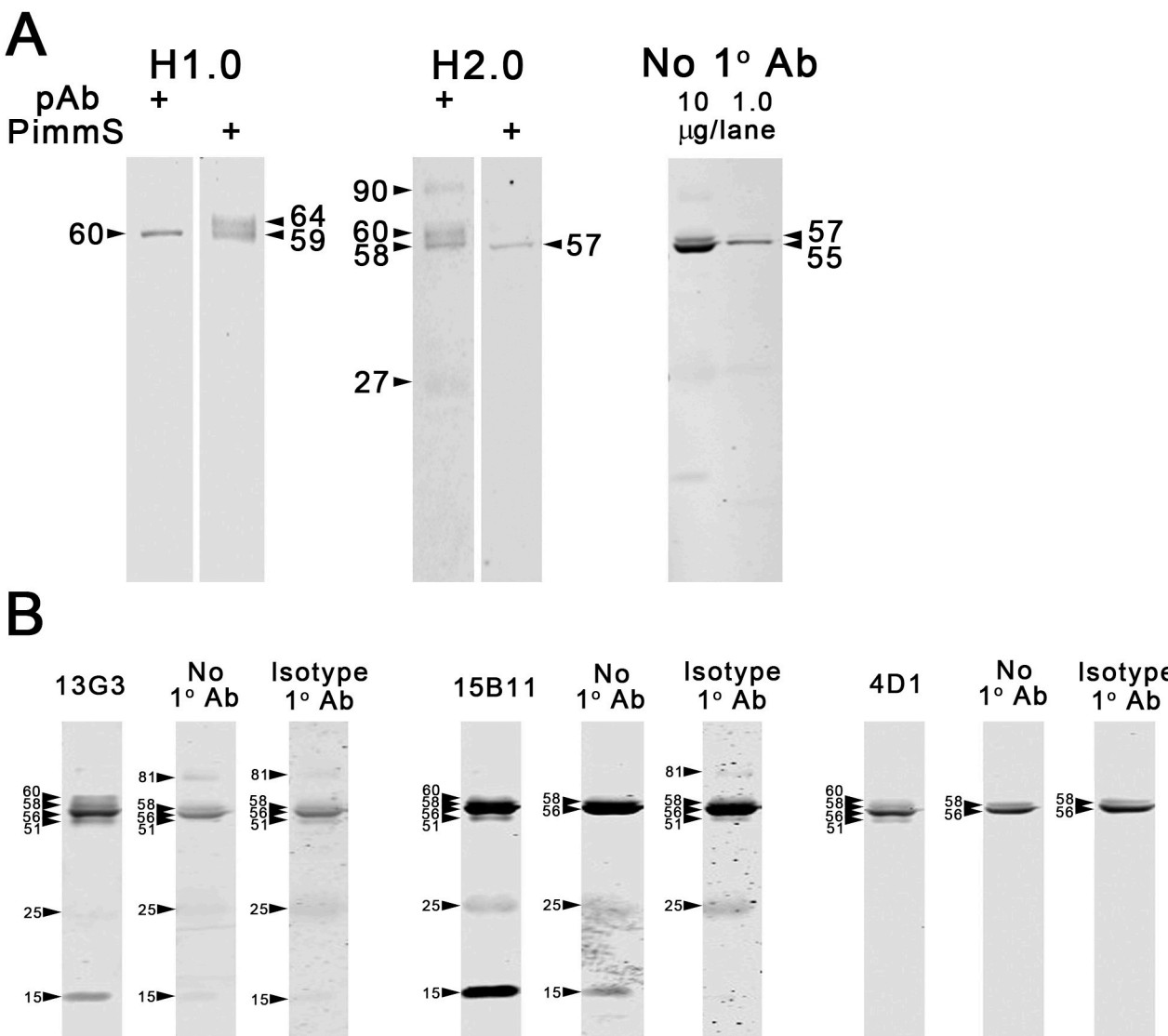

**Fig 5. Western blot analyses of CABS1 in human (male) saliva.** A) In this experiment (pooled human saliva sample, n = 1, 2 to 4 independent replicates), optimal specificity with pAb and preimmune serum (PimmS) was achieved at 2.2 μg of human saliva and 0.2 to 0.3 ng/μL of pAb and preimmune sera and no primary antibody control. B) For mAb 13G3 10 ng/μL and for mAb 15B11 and 4D1 1 ng/μL were used together with 10 μg of saliva (n = 3 for each mAb). Controls with no primary mAb or with isotype control mAb are shown.

experiments showed a specific immunoreactive band(s) at ~61 in comparison to isotype controls (S2 Fig). No signal was present when probing the membrane only with secondary antibody.

In summary, specific immunoreactive CABS1 was detected in serum at ~ 61kDa with mAb.

## Immunohistochemical analysis of human testis and SMG

Because the only previous immunohistochemical studies of CABS1 were done with testes of mice [19] and rats [20] and subsequently with pig [21], we conducted immunoperoxidase

studies of hCABS1 in human testes for comparison. Thereafter, we investigated hCABS1 expression by immunoperoxidase and immunofluorescence in human SMG.

**Testes.**   Fig 6A, 6C and 6E show the normal morphology of a human testis stained with H&E. Seminiferous tubules (ST), interstitial Leydig cells (LC) and, within the tubules, primary

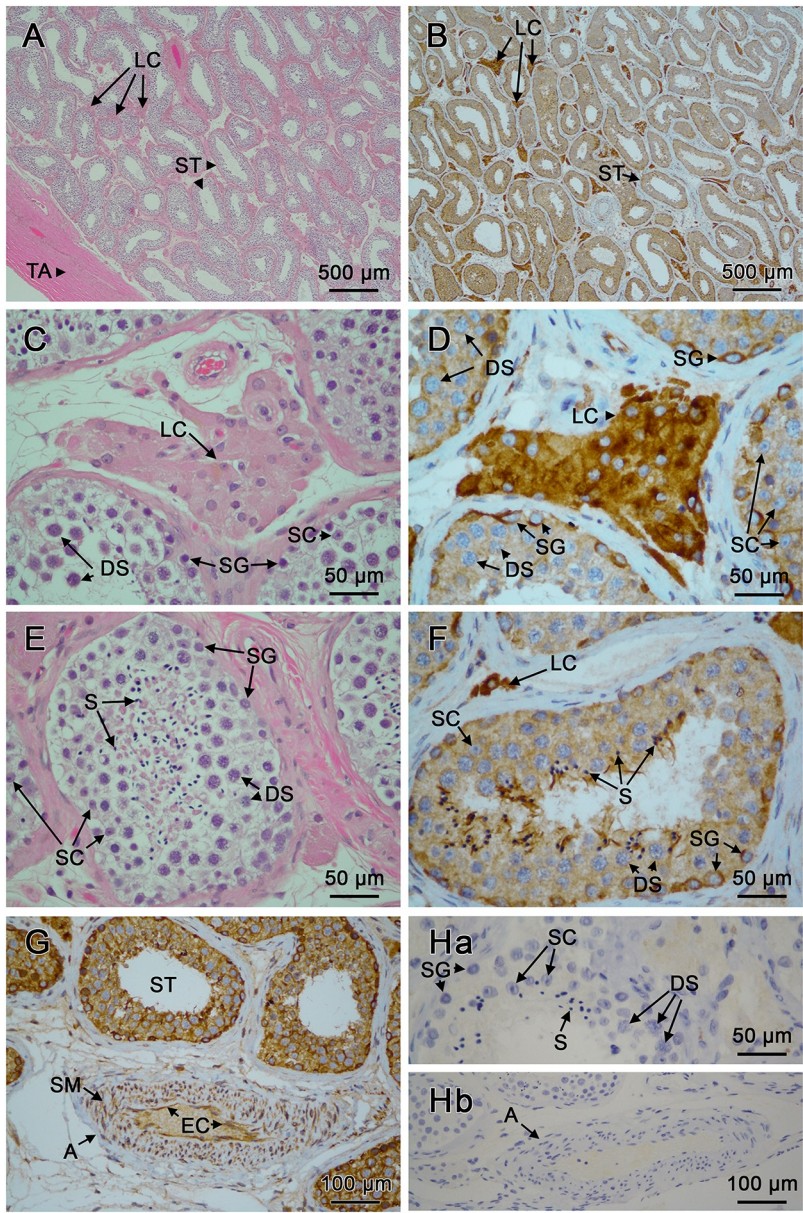

**Fig 6. Immunohistochemical analyses of CABS1 in human testes.** A) Low power photomicrograph of the connective tissue of the tunica albuginea (TA), seminiferous tubules (ST), and Leydig cells (LC) stained with Hematoxylin & Eosin (H&E). B) Low power photomicrograph of ST and LC stained with 15B11 monoclonal antibody to human CABS1. C) H&E staining of LC and, in ST, primary spermatogonia (SG), developing spermatocytes (DS) and supporting Sertoli cells (SC) (prominent nucleoli). D) Similar image to C, stained with 15B11; note strong staining of LC, and also staining of SG, DS and SC. E) H&E stain of the cell types in ST, including spermatids (S). F) CABS1 staining observed in seminiferous tubule showing positivity in SG, DS and S G) CABS1 staining including most intensely stained SG, with less intense staining in DS and SC; luminal spermatids show variable staining. An adjacent artery (A) also exhibits staining in the vascular smooth muscle (SM) and endothelium cells (EC). Ha) Negative control (secondary antibody only; isotype controls were also negative) in cells of seminiferous tubules (upper) and (Hb) an adjacent artery. Bars on each figure represent size in microns.

spermatogonia (SG), Sertoli cells (SC), developing spermatocytes (DS) and spermatids (S) can be seen (labels). Immunoreactivity using mAb 15B11 detected hCABS1 in LC in the interstitial tissue of the testis (Fig 6B, 6D and 6F), and in SG in ST (Fig 6D, 6E and 6H). In some ST developing spermatids (S) were also positive for CABS1 immunoreactivity (Fig 6F). Controls done with no primary antibody or with isotype control antibody were negative (Fig 6F top and lower panel).

**Submandibular glands.** Normal morphology of the SMG is shown in Fig 7A and 7D at increasing magnification. The lobular structure of the gland can be seen, as well as interlobular connective tissue containing vasculature, and excretory ducts (ED). Within the lobes (L), serous (SA) and mucous acini (MA) are evident, together with numerous smaller salivary ducts (SD) in different planes of section. Using mAb 15B11, salivary ducts of all sizes had hCABS1 immunoreactivity in epithelial cells (Fig 7B, 7E, 7G,7H and 7J) Controls with no primary antibody or with isotype control antibodies were negative for immunoreactivity throughout the SMG (Fig 7C, 7F, 7I and 7L). SA cells were positive and the basal, paranuclear regions of MA cells were positive for hCABS1 (Fig 7E); the mucous-containing regions of these cells were negative for hCABS1. Interestingly, nerves (N) (Fig 7J and 7K) were hCABS1 positive and some vessels had hCABS1 immunoreactivity in endothelial cells (EC, Fig 7K and 7F shows the negative control for EC).

Immunofluorescence (see Methods) staining of human submandibular gland sections with pAbs H1.0 and H2.1 yielded a punctate pattern throughout the cytoplasm of duct cells (Fig 7M). Interestingly, pAbs H2.0 and H2.2 exhibited strong CABS1 immunoreactivity in the cytoplasm of specific basal (abluminal, AB) epithelial duct cells (Fig 7N). Analyses of SMGs immunostained with preimmune sera for the pAb gave no immunoreactivity (Fig 7O). Additional figures of our immunofluorescence studies are available in S3.4–S3.8 Fig. Moreover, as with immunoperoxidase studies above, in immunofluorescence studies SA and MA were positive (S3.1 and S3.2 Fig), as were EC (S3.5 Fig). No obvious differences were detected in hCABS1 protein expression in SMG resected from Sjogren's or cancer subjects (grossly normal tissues selected, see Discussion).

## GEOdata assessment of CABS1 tissue distribution and relative abundance

In our search of hCABS1 expression in different tissues in public domain transcriptome databases, we selected three papers that surveyed numerous normal human tissues for gene expression [25–27]. Dezso et al (2008) [26] used the Applied Biosystems human genome microarray of 27,868 genes and 31 normal tissues; She et al (2009) [25] used oligonucleotide arrays from Agilent Technologies Inc, detected 18,149 genes from 42 normal human tissues; Wang et al 2019 [27] used the Illumina HiSeq 2000/2500 system and detected 18,702 protein-encoding genes from 29 normal human tissues. Fig 8 summarizes the normalized relative abundance of hCABS1 from the transcriptomic data from these three studies. CABS1 expression in testes is the best studied of the tissues, but CABS1 mRNA has been found throughout the urogenital, gastrointestinal and respiratory tracts, as well as in glands, the nervous system, immune system and other sites.

## Discussion

To validate and extend our previous studies with WB [18, 22] and NCIA [24], we described studies of hCABS1 in transient overexpression lysate (OEL), SMG lysates, saliva, serum and testis using pAb and mAb. WB and immunohistochemistry studies were used together with GEOdata analyses to validate our observations of hCABS1 in these tissues and fluids.

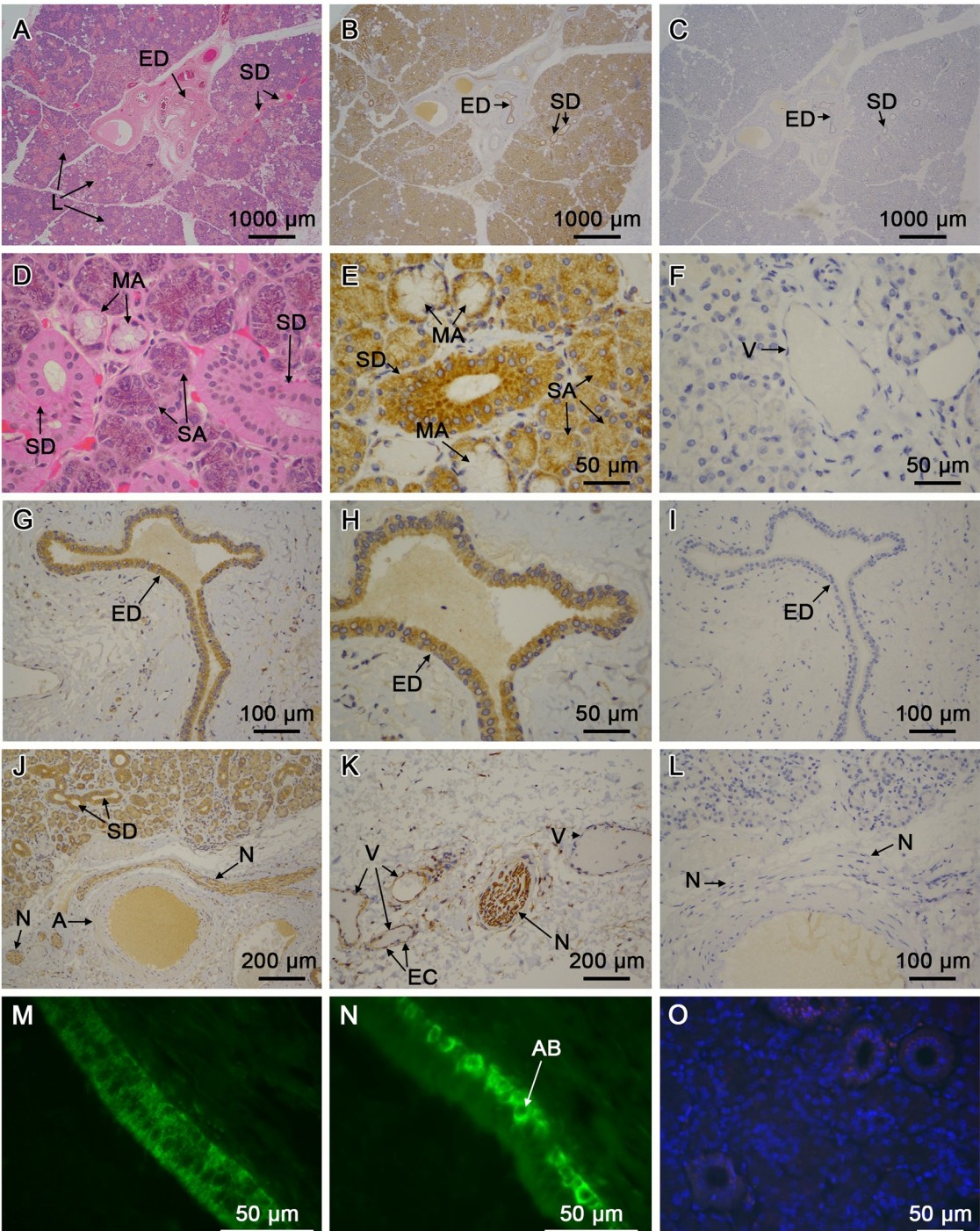

**Fig 7. Immunohistochemical analyses of CABS1 in human submandibular glands.** A) H&E staining showing lobules (L) of the gland together with the excretory salivary duct (ED) in interlobular connective tissue. Within the glandular tissue of the L there are sections of smaller collecting salivary ducts (SD) among acini (see below) and associated vasculature. B) Similar section showing CABS1 staining of ducts and acini as well as nerve (N). C) Similar section viewed with negative control staining (no primary antibody; isotype controls were also negative). D) H&E staining of serous (SA) and mucous acini (MA) and associated SD. E) Tissue section similar to D) with CABS1 staining of SD, SA and MA. Mucin is negative. F) Negative control showing acini and thin-walled vessel (V), consistent with a capillary. G) CABS1 staining in ED. H) Higher magnification of CABS1 staining in ED epithelium. I) Negative control for CABS1 staining in the ED. J) CABS1 staining in nerves (N) within the salivary gland. Also shown is an artery (A), and SD. K) CABS1 staining in N as well as in V endothelium. L) Negative control view (no primary antibody, isotype control negative as well). M and N) Selective immunofluorescent staining of CABS1 (rabbit polyclonal antibodies H2.1 and H2.2,

respectively). H2.2 and H2.0 (see also S3.4–S3.8 Fig) showed selective cellular cytoplasmic staining abluminal cells (AB) in the basal regions of the epithelium of ED, whereas H2.1 stained the cytoplasm of cells throughout the epithelium. Controls with preimmune sera (e.g. H2.2, O) or no primary antibody were negative. Bars on each figure represent size in microns.

The results of WB studies of OEL conducted with pAb and mAb as well as anti-FLAG antibody are summarized in Table 1. Our previous results using NCIA [24] are also included for comparison, as well as the distribution of CABS1 peptides, determined by MS-seq in various sections ($M_r$ ranges) of a 1 D gel (Table 1). $M_r$ estimates of 84 and 78 are likely to be the same immunoreactive band given time differences between studies with pAb and mAb, other potential technical differences, and methods of $M_r$ calculation. Indeed, results with anti-FLAG antibody confirm that 84 (pAb) = 78 (mAb) (see Fig 2B, 4D1).

For WB studies, estimated $M_r$ values are for specific immunoreactive bands detected in OEL; no hCABS1 specific bands were detected in NCL (normal control lysate). Several commercial batches of OEL lysate were used (n = 7), and it is possible (but cannot be confirmed) that some of the differences in $M_r$ between results with pAb and mAb could reflect differences among batches of OEL. $M_r$ estimates for pAb studies were derived from LI-COR software, whereas for mAb studies estimates were from manual calculations, as the LI-COR software was no longer available for our use. The commercial supplier of NCIA Wes™ estimates that given their separation matrix, estimates of $M_r$ may vary by up to 20% from those derived from SDS PAGE separations [24]. The anti-FLAG results were positive controls run with either the pAb or mAb WB studies and involved a single source of mAb anti-FLAG.

MS shows CABS1 fragments in six segments of the 1 D gel ranging from 119–85 kDa to 16–11 kDa. Given the peptides identified in gel segments, there is no evidence of selective

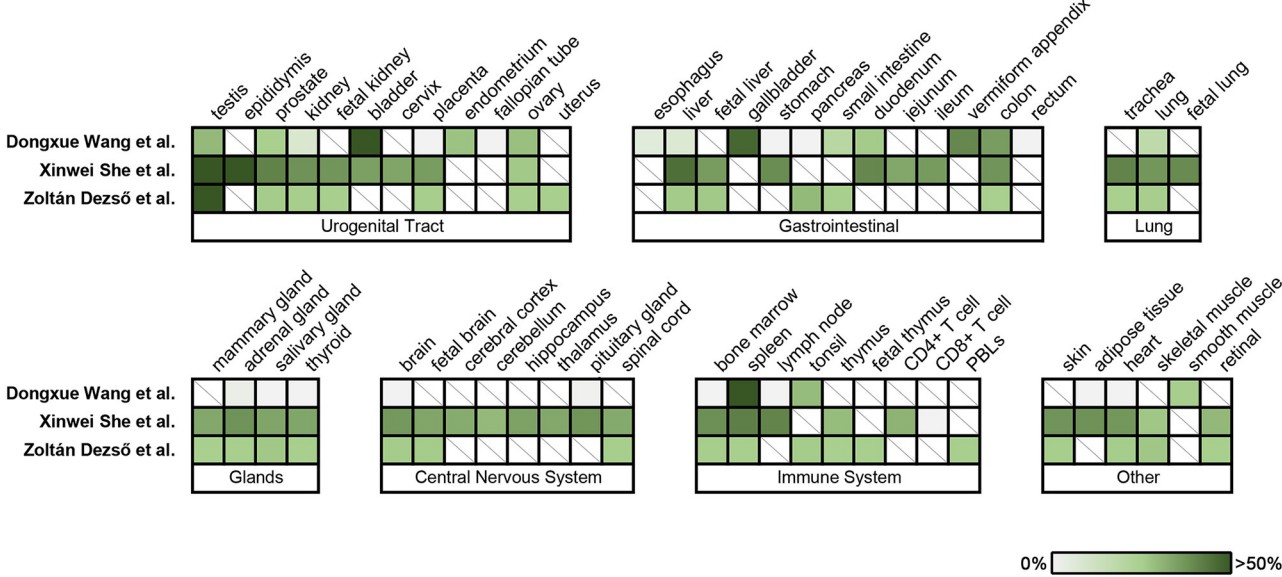

**Fig 8. Meta-analysis of the distribution of CABS1 in normal human tissue across three studies.** Each study is represented in a row, and tissues are represented in columns. Tissues are grouped in their respective biological system. CABS1 GEOdata was extracted and normalized by dividing each expression read by the maximum value observed across tissues within the same study. The normalized percentage of CABS1 expression is shown on a continuous graded scale ranging from 0% (grey) to more than 50% (green), as indicated by the color key. A diagonal line in a box indicates the lack of data within that tissue.

**Table 1. Estimated $M_r$ of CABS1 detected in transient overexpression lysate (OEL) by pAb and mAb in WB and nano-capillary immunoassay (NCIA) and by mass spectroscopy (MS).**

| Technique | Antibody Name | $M_r$ (kDa) |
|---|---|---|
| WB pAb | H1.0 | 84, 67 |
| | H2.0 | 84, 67 |
| | H2.1 | 84 |
| | H2.2 | 84, 67 |
| | Anti-Flag (+ve Control) | 84, 67 |
| MS | | 119–85, 84–60, 59–40, 39–33, 32–22, 16–11 |
| WB mAb | 13G3 | 78 |
| | 15B11 | 78 |
| | 4D1 | 78 |
| | Anti-Flag (+ve Control) | 78 |
| NCIA | H1.0 | 94, 65 |
| | H2.0 | 94, 65 |

fragmentation, i.e., existence of specific portions of CABS1 in distinct gel fragments (data repository folder, mass spectroscopy section).

We have previously published our results with pAbs on immunoreactive bands of hCABS1 in SMG lysates [18]. Multiple bands were detected with H2.0 and other pAb, including: bands at 51, 33, 27, and, in some SMG lysates, at 20, 17, 16 and 11 kDa. In the present study with SMG lysate and using H1.0, 2.0, 2.1, and 2.2 anti-CABS1 pAbs, specific immunoreactive bands were detected with at least three of four pAbs at ~71, 52 and 27 kDa (Fig 4A; Table 2).

With at least two of three anti-CABS1 mAbs, immunoreactive bands were found in SMG lysate at 63 and 53 kDa (Fig 4B, Table 2). Human CABS1 immunoreactive bands were detected using pAbs and mAbs in saliva at similar molecular sizes, 60 to 63 and 52 to 53 kDa, results consistent with our earlier report using NCIA [24] (Table 2). Moreover, using mass spectroscopy, rhCABS1 was detected in OEL at 120–85, 84–60, 59–40, 39–22 and 16–11 kDa (Table 1). Two of three mAbs identified a specific immunoreactive band in serum at ~61 kDa (S2 Fig). However, with four pAbs numerous bands were detected in a serum pool and no obvious reproducible profile could be identified. Interestingly, CABS1 has been detected in human serum by others using mass spectroscopy, although only a single peptide was identified that covered 3.54% of the protein [28].

With the newly optimized antigen and pAb concentrations, in contrast to previous work [18, 22], we only detected a faint specific band at ~27 kDa in saliva (Table 2). However, a 27

**Table 2. Molecular size estimates of human-specific CABS1 immunoreactivity (polyclonal and monoclonal antibodies) in submandibular gland lysate, saliva and serum*.**

| Polyclonal Ab | H1.0 | | H2.0 | | H2.1 | | H2.2 |
|---|---|---|---|---|---|---|---|
| Monoclonal Ab | | 13G3 | | 15B11 | | 4D1 | |
| Submandibular Gland Lysate | 52, 27 | 63, 53 | 71, 52, 27 | 53 | 71, 52, 27 | 63, 53 | 71, 52 |
| Saliva | 60 | 60 | 60 | 60 | NA | 60 | NA |
| NCIA (Saliva) | 60 | NA | 58, 34 | NA | NA | NA | NA |
| Serum[#] | NA | 61 | NA | ND | NA | 61 | NA |

*Summary from Figs 4 and 5, and S2 Fig.

[#]Pilot study (n = 1); multiple bands, no obvious consistency with pAb, labeled Not Applicable (NA). Not detected bands were labeled Not Detected (ND).

kDa band was detected in SMG lysate, in mass spectroscopy of OEL (32–22 kDa gel fragment) and in NCIA (34 kDa given the different gel matrices used, allowing for up to 20% differences in $M_r$ estimation between polyacrylamide gels and NCIA, see above).

The challenge in detecting a strong 27 kDa band in human saliva using our newly optimized pAb protocol, and mAb, may be explained by low abundance of this form of the protein. Mass spectroscopic analysis of the entire full gel electrophoresis column of OEL showed that the highest CABS1 expression levels were for molecules in the 84–60 kDa range (Fig 3). The second highest detected range was 6 times lower expression levels in the 39–33 kDa range. Although CABS1 peptides were detected at lower molecular weights, the abundance decreased to 19 times lower for the 32–22 kDa range. The concentration of saliva (2.2 μg/well) and pAb (0.3 ng/μL) that yielded highest specificity in WB profiles in our newly optimized protocol was much lower than concentrations we used in previous pAb studies (saliva, 25 μg; pAb 3 ng/μL) [18]. A combination of lower saliva sample concentration, lower pAb and mAb concentrations, and low relative abundance of 27 kDa CABS1 could explain our ability to detect only a faint band of this potential form of CABS1 in saliva. However, it is also possible that our earlier results [18] were incorrect. Thus, we are unable to confirm the presence of a relatively abundant 27 kDa CABS1 form in human saliva. This raises concerns about whether the 27 kDa molecular form that we reported to be associated with acute stress is indeed hCABS1; a question that remains pending until new studies are conducted and replicated by others. Regrettably, we have been unable to sequence hCABS1 from human SMG and saliva using mass spectroscopy, possibly because of its low abundance proportionate to other proteins. However, our immunohistochemical and GEOdata studies confirm the presence of CABS1 in human SMG (see below).

A limitation of our studies is that they were conducted over 6 years with several pAb and mAb and with several samples of OEL and NCL (n = 7 to 8), and SMG lysates (n = 8, S2 Table). For serum and saliva, care was taken to create sample pools that were aliquoted and frozen (-80°C), and not repeatedly frozen and thawed. WB optimization protocols varied among antibodies and samples but were established with careful titrations. Moreover, although $M_r$ standards were routinely used, the methods for identification of $M_r$ of immunoreactive bands varied; initially this was done using the LI-COR software, but beginning in 2019 molecular weights were calculated manually, as the LI-COR software was no longer available to us. These many issues likely influenced the variability encountered in $M_r$ estimates using WB. Furthermore, it was well-recognized that CABS1 is a structurally disordered protein with conformational plasticity [19, 20, 29–31] and variable estimates of $M_r$ of the putatively mature protein (66 to 79 kDa), considerably different from that predicted for the newly translated polypeptide (42 kDa). This flexible, disordered structure, although potentially of functional significance, may also contribute to the variability in $M_r$ that we detected among the tissues and fluids studied. Another component of the variability in $M_r$ for hCABS1 may involve the cell of its origin in the tissues (e.g., spermatogonia, epithelium, nerves, etc.), cell type-specific processing and CABS1 interactions with charged ligands [29].

As previous literature on the cellular localization of CABS1 protein focused on testes in rodents [19, 20] and pigs [21], we conducted the first immunohistochemical cellular localization of CABS1 in human testes (Fig 6). An earlier study had indicated that CABS1 mRNA was abundant in human testes [32]. We found CABS1 in developing DS, S, in human ST (Fig 6). Interestingly, unlike previous rodent and porcine studies, CABS1 was also found in SG, SC and LC in human testes. In mice [19] and rats [20] CABS1 was found in late phases of spermatid development, including in the inner mitochondrial membrane in rats [20] and in the flagellum in mature sperm [19]. In the pig, CABS1 was found in elongated spermatids and sperm, specifically in the principal and end piece of the flagellum, as well as in the acrosome [22].

Interestingly, the shedding of the porcine acrosome (acrosomal reaction) includes the loss of CABS1, and the acrosomal reaction can be inhibited by anti-CABS1 antibodies, without an effect on sperm viability [22]. The knockdown of CABS1 in mice disrupted sperm tail structure, induced an abnormality in the flagellum, specifically at the midpiece-principal region junction, and reduced fertility [33]. Taken together this evidence from rat, mouse and pig suggests that CABS1 is involved in spermatogenesis, sperm motility and the acrosomal reaction. Given the presence of CABS1 in human SG, SC and LC, but not these locations in mice, rats and pigs, CABS1 may play additional roles in male reproduction in humans, perhaps in association with the unique expression of a peptide sequence near the carboxyl terminus in selected primates [29] that has anti-inflammatory activity [18].

There are few reports of hCABS1 in the human female urogenital tract [34, 35], although as shown in Fig 8, there are numerous reports of CABS1 mRNA. In transcriptomic [25–27] analyses of healthy human tissues, hCABS1 was identified in endometrium, ovary, fallopian tubes, placenta, cervix and urinary bladder. Cerny et al. [34] showed in bovine oviductal epithelial cells that levels of CABS1 transcript changed during the estrous cycle. Moreover, Calhoun et al. [35] showed that exposure to the toxin bisphenol A during fetal development altered CABS1 expression in the fetal uterus and suggested that this may influence uterine function later in life. Clearly, much needs to be learned about the roles of CABS1 in reproductive physiology.

Interestingly, our immunohistochemical studies detected human CABS1 in several anatomical compartments and cell types of SMG, including epithelial cells of SA, MA and SD and ED. Connecting hCABS1 to a potential anti-inflammatory role in humans, it has been reported that in Sjogren's syndrome, an autoimmune inflammation of the salivary glands, CABS1 mRNA levels were decreased 5.4-fold in comparison to normal [36]. This reduction in hCABS1 mRNA in Sjogren's syndrome was confirmed with laser capture microdissection of the acinar and ductal epithelium of minor labial salivary glands [37]. Our immunofluorescence studies did not detect obvious differences in CABS1 abundance between SMG from Sjogren's subjects or normal tissue regions from surgical resections for cancer.

Our observations that nerves and vascular endothelium and smooth muscle in SMG are also CABS1 positive are novel observations, further supported by GEOdata sets that identify CABS1 in SMG, nerves, endothelium and smooth muscle [25–27]. These studies also suggest that hCABS1 is widely distributed in many other tissues including liver, gastrointestinal tract, respiratory tract, neuro, endocrine and immune systems. Given the structural plasticity of CABS1, its widespread tissue distribution and apparent functional diversity, e.g., from spermatogenesis to anti-inflammatory activities, it is imperative to identify the specific cell types that express CABS1 and its functions.

## Supporting information

**S1 Fig. Binding of mAb 4D1 to peptide 3 (AA 40–60) of CABS1.**
(PDF)

**S2 Fig. Western blot analyses of CABS1 in human (male) serum.**
(TIF)

**S3 Fig. Immunofluorescence studies of submandibular glands.**
(PDF)

**S1 Table. Glossary of abbreviations.**
(PDF)

**S2 Table. Human testes and submandibular gland samples.**
(PDF)

**S1 File.**
(PDF)

**S2 File.**
(PDF)

**S3 File.**
(PDF)

**S4 File.**
(PDF)

**S5 File.**
(PDF)

**S6 File.**
(DOCX)

## Acknowledgments

We wish to thank Sarah Canill, Alberta Precision Laboratories, and Yong-Qiu Doughman, Case Western Reserve University for assistance with immunohistochemical studies. Mr. Cameron Lloyd, Anatomical Pathology, University of Alberta Hospital was instrumental in identification of archived tissue specimens. Mr. Jack Moore and Dr. Richard Fahlman of The Alberta Proteomics and Mass Spectroscopy Facilities were invaluable in the mass spectroscopy analysis of samples. Christopher St. Laurent, University of Alberta, assisted with western blot analyses. ER-S received graduate studentships from the Faculties of Graduate Studies and Research, and Medicine and Dentistry, University of Alberta. MM-P received funding from the Natural Sciences and Engineering Research Council, Canada. ADB received funding from AllerGen NCE Inc, Canada.

## Author Contributions

**Conceptualization:** Eduardo Reyes-Serratos, Joy Ramielle L. Santos, Aron Gonshor, Robert Buck, A. Dean Befus, Marcelo Marcet-Palacios.

**Data curation:** Eduardo Reyes-Serratos, Joy Ramielle L. Santos, Lakshmi Puttagunta, Mechiko Watanabe, Robert Buck, A. Dean Befus, Marcelo Marcet-Palacios.

**Formal analysis:** Eduardo Reyes-Serratos, Joy Ramielle L. Santos, Lakshmi Puttagunta, Mechiko Watanabe, Robert Buck, A. Dean Befus, Marcelo Marcet-Palacios.

**Funding acquisition:** A. Dean Befus, Marcelo Marcet-Palacios.

**Investigation:** Eduardo Reyes-Serratos, Joy Ramielle L. Santos, Lakshmi Puttagunta, Mechiko Watanabe, Robert Buck, A. Dean Befus, Marcelo Marcet-Palacios.

**Methodology:** Eduardo Reyes-Serratos, Joy Ramielle L. Santos, Lakshmi Puttagunta, Mechiko Watanabe, Aron Gonshor, Robert Buck, A. Dean Befus, Marcelo Marcet-Palacios.

**Project administration:** Lakshmi Puttagunta, Mechiko Watanabe, Aron Gonshor, Robert Buck, A. Dean Befus, Marcelo Marcet-Palacios.

**Resources:** Lakshmi Puttagunta, Stephen J. Lewis, Mechiko Watanabe, Aron Gonshor, Robert Buck, A. Dean Befus, Marcelo Marcet-Palacios.

**Software:** Eduardo Reyes-Serratos, Joy Ramielle L. Santos, Lakshmi Puttagunta, A. Dean Befus, Marcelo Marcet-Palacios.

**Supervision:** Mechiko Watanabe, A. Dean Befus, Marcelo Marcet-Palacios.

**Validation:** Eduardo Reyes-Serratos, Joy Ramielle L. Santos, Lakshmi Puttagunta, Stephen J. Lewis, Mechiko Watanabe, Aron Gonshor, Robert Buck, A. Dean Befus, Marcelo Marcet-Palacios.

**Visualization:** Eduardo Reyes-Serratos, Joy Ramielle L. Santos, Lakshmi Puttagunta, Mechiko Watanabe, Aron Gonshor, Robert Buck, A. Dean Befus, Marcelo Marcet-Palacios.

**Writing – original draft:** Eduardo Reyes-Serratos, Joy Ramielle L. Santos, Lakshmi Puttagunta, A. Dean Befus, Marcelo Marcet-Palacios.

**Writing – review & editing:** Eduardo Reyes-Serratos, Joy Ramielle L. Santos, Lakshmi Puttagunta, Stephen J. Lewis, Mechiko Watanabe, Aron Gonshor, Robert Buck, A. Dean Befus, Marcelo Marcet-Palacios.

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
