## [Decision Letter · Decision Letter 0]

24 Oct 2023

PONE-D-23-29960Identification and Characterization of Calcium Binding Protein, Spermatid Associated 1 (CABS1) in Selected Human Tissues and FluidsPLOS ONE

Dear Dr. Marcet-Palacios,  

Thank you for submitting your manuscript to PLOS ONE. After careful consideration, we feel that it has merit but does not fully meet PLOS ONE’s publication criteria as it currently stands. Therefore, we invite you to submit a revised version of the manuscript that addresses the points raised during the review process.

We look forward to receiving your revised manuscript.

Kind regards,

Abeer El Wakil, PhD

Academic Editor

PLOS ONE

Journal Requirements:

1. When submitting your revision, we need you to address these additional requirements. Please ensure that your manuscript meets PLOS ONE's style requirements, including those for file naming. The PLOS ONE style templates can be found at https://journals.plos.org/plosone/s/file?id=wjVg/PLOSOne_formatting_sample_main_body.pdf and https://journals.plos.org/plosone/s/file?id=ba62/PLOSOne_formatting_sample_title_authors_affiliations.pdf 2. Thank you for stating the following financial disclosure: MMP, RGPIN-2020-04553, Natural Sciences and Engineering Research Council ofCanada, https://www.nserc-crsng.gc.ca/.ADB, 16BB-MSI-C5 and 18BB-SI-C9. AllerGen. https://www.allergen.ca/. Please state what role the funders took in the study.  If the funders had no role, please state: "The funders had no role in study design, data collection and analysis, decision to publish, or preparation of the manuscript." If this statement is not correct you must amend it as needed. Please include this amended Role of Funder statement in your cover letter; we will change the online submission form on your behalf. 3. Thank you for stating the following in the Competing Interests section: I have read the journal's policy and the authors of this manuscript have the following competing interests: * A. Dean Befus, together with the University of Alberta holds a patent on CABS1 as a biomarker and an associated licensing agreement with GB Diagnostics.* Aron Gonshor and Robert Buck are co-owners of GB Diagnostics.  We note that one or more of the authors are employed by a commercial company: name of commercial company.  a. Please provide an amended Funding Statement declaring this commercial affiliation, as well as a statement regarding the Role of Funders in your study. If the funding organization did not play a role in the study design, data collection and analysis, decision to publish, or preparation of the manuscript and only provided financial support in the form of authors' salaries and/or research materials, please review your statements relating to the author contributions, and ensure you have specifically and accurately indicated the role(s) that these authors had in your study. You can update author roles in the Author Contributions section of the online submission form. Please also include the following statement within your amended Funding Statement. “The funder provided support in the form of salaries for authors, but did not have any additional role in the study design, data collection and analysis, decision to publish, or preparation of the manuscript. The specific roles of these authors are articulated in the ‘author contributions’ section.”If your commercial affiliation did play a role in your study, please state and explain this role within your updated Funding Statement.  b. Please also provide an updated Competing Interests Statement declaring this commercial affiliation along with any other relevant declarations relating to employment, consultancy, patents, products in development, or marketed products, etc.   Within your Competing Interests Statement, please confirm that this commercial affiliation does not alter your adherence to all PLOS ONE policies on sharing data and materials by including the following statement: "This does not alter our adherence to  PLOS ONE policies on sharing data and materials.” (as detailed online in our guide for authors http://journals.plos.org/plosone/s/competing-interests) . If this adherence statement is not accurate and  there are restrictions on sharing of data and/or materials, please state these. Please note that we cannot proceed with consideration of your article until this information has been declared. Please include both an updated Funding Statement and Competing Interests Statement in your cover letter. We will change the online submission form on your behalf.
 4. In your Data Availability statement, you have not specified where the minimal data set underlying the results described in your manuscript can be found. PLOS defines a study's minimal data set as the underlying data used to reach the conclusions drawn in the manuscript and any additional data required to replicate the reported study findings in their entirety. All PLOS journals require that the minimal data set be made fully available. For more information about our data policy, please see http://journals.plos.org/plosone/s/data-availability. Upon re-submitting your revised manuscript, please upload your study’s minimal underlying data set as either Supporting Information files or to a stable, public repository and include the relevant URLs, DOIs, or accession numbers within your revised cover letter. For a list of acceptable repositories, please see http://journals.plos.org/plosone/s/data-availability#loc-recommended-repositories. Any potentially identifying patient information must be fully anonymized. Important: If there are ethical or legal restrictions to sharing your data publicly, please explain these restrictions in detail. Please see our guidelines for more information on what we consider unacceptable restrictions to publicly sharing data: http://journals.plos.org/plosone/s/data-availability#loc-unacceptable-data-access-restrictions. Note that it is not acceptable for the authors to be the sole named individuals responsible for ensuring data access. We will update your Data Availability statement to reflect the information you provide in your cover letter.  5. Please include a caption for figure 4. 6. Please include captions for your Supporting Information files at the end of your manuscript, and update any in-text citations to match accordingly. Please see our Supporting Information guidelines for more information: http://journals.plos.org/plosone/s/supporting-information.  7. PLOS ONE now requires that authors provide the original uncropped and unadjusted images underlying all blot or gel results reported in a submission’s figures or Supporting Information files. This policy and the journal’s other requirements for blot/gel reporting and figure preparation are described in detail at https://journals.plos.org/plosone/s/figures#loc-blot-and-gel-reporting-requirements and https://journals.plos.org/plosone/s/figures#loc-preparing-figures-from-image-files. When you submit your revised manuscript, please ensure that your figures adhere fully to these guidelines and provide the original underlying images for all blot or gel data reported in your submission. See the following link for instructions on providing the original image data: https://journals.plos.org/plosone/s/figures#loc-original-images-for-blots-and-gels.   In your cover letter, please note whether your blot/gel image data are in Supporting Information or posted at a public data repository, provide the repository URL if relevant, and provide specific details as to which raw blot/gel images, if any, are not available. Email us at plosone@plos.org if you have any questions.

Additional Editor Comments:

Dear Authors,

The concept of the present study seems interesting. In this work, the authors address an interesting and original topic, which is the characterization CABS1 by using different approaches (immunohistochemistry, WB and bioinformatics) and it moves from the development of a technique just validated to obtain monoclonal and polyclonal antibodies. However, the manuscript in its actual form does not meet the standards set forth by Plos One. I highly recommend the authors to address precisely the reviewers’ concerns before delivering any decision.

Thank you,

Abeer El Wakil.

Reviewers' comments:

Reviewer's Responses to Questions

**Comments to the Author**

1. Is the manuscript technically sound, and do the data support the conclusions?

Reviewer #1: Yes

Reviewer #2: Partly

2. Has the statistical analysis been performed appropriately and rigorously? 

Reviewer #1: Yes

Reviewer #2: No

3. Have the authors made all data underlying the findings in their manuscript fully available?

Reviewer #1: Yes

Reviewer #2: Yes

4. Is the manuscript presented in an intelligible fashion and written in standard English?

Reviewer #1: Yes

Reviewer #2: Yes

5. Review Comments to the Author

Reviewer #1: Comments about the manuscript:

“Identification and Characterization of Calcium Binding Protein, Spermatid Associated 1 (CABS1) in Selected Human Tissues and Fluids”

Calcium-binding protein associated with spermatid 1 (CABS1) is widely studied in spermatogenesis. Its mRNA is present in many other tissues, but with little information about it. The aim of the study presented here was to investigate the presence of this protein in different tissues (salivary glands, saliva, serum and testes) using polyclonal and monoclonal antibodies directed against different sections of the protein, prepared during from previous work, by Western blot, immunohistochemistry and bioinformatics.

This substantial work brings interesting and useful results. The manuscript nevertheless needs to be improved before considering its publication.

Page 5 “WB”: western blot? Write in full the first time the name appears in the text.

This remark is applicable throughout the manuscript. Please check this point. On the other hand, when the number of abbreviations is large (like here), it would be useful to provide a glossary of abbreviations.

Page 6. “Post-homogenization, samples were centrifuged”: specify the characteristics of the centrifugation: Speed (in number of g), duration.

Page 9. Immunohistochemical analyses. In the description of the technique, specify how the negative controls were carried out.

Page 9. “SMG“: write in full SMG (submandibular gland, I think?) with abbreviation between brackets, when it is encountered for the first time in the text. See note above.

Page 12, Figure 1 - Western blot analyses of human CABS1 in transient overexpression lysate (OEL) of HEK 293T cells and in control lysate (NCL) of HEK 293T cells transfected with empty plasmid: Is this really figure 1? it looks to me like figure 2. Please check and correct.

Page 12, Figure 2 – Estimate of the abundance of hrCABS1 in sections (range of kDa) of 1 D gel analyzed by mass spectroscopy: same: Is it not figure 3? Please, check and correct.

Page 13, Figure 3 – Western blot analyses of CABS1 in human female and male submandibular gland lysates. A: Is it not figure 4? Please check and correct.

Between Figures 3 and 5 (Figure 5 – Western blot analyzes of CABS1 in human (male) saliva), there is no Figure 4. Please check.

Page 15, at the bottom of the page, “Figure 5 – Western blot analyses of CABS1 in human (male) saliva.”: this figure has already been given in the text.

Page 17. Figure 7 – Immunohistochemical analyses of CABS1 in human testes. “(lower, 200X)”: use a scale bar instead of magnification because magnification varies depending on the size of the text sheet.

Page 18, line 1. “antibody or with isotype control antibodies were negative3: Specify the controls on material and methods (see comments).

Page 18, Figure 8 - Immunohistochemical analyses of CABS1 in human submandibular glands: use a scale bar on the pictures, and not a magnification (see remark above).

In the legend, “striated tubes“ are designed with “DD”, but with “SD” on the picture.

No abbreviation is associated to “excretory duct” in the legend. It is ED on the picture.

Reviewer #2: Dear Authors, the project idea of your manuscript is very ambitious because it aims to characterize CABS1 by using different approaches (immunohistochemistry, WB and bioinformatics) and it moves from the development of a laborious technique to obtain monoclonal and polyclonal antibodies. Albeit the study design seems well conceived considering the wide typology of human samples (tissues and fluid) to examine for the identification of CABS1, several flaws are present. The manuscript in fact is well described for the chapter of polyclonal and monoclonal antibodies production as well as for the mass spectrometry sequencing. Differently, the section of the manuscript referring to CABS1 characterization at level of tissues and fluids has scarce scientific basis considering the limited quantity of examined sample (human testis, blood and saliva), as well as an incomplete description of materials and method (western blotting). To demonstrate the expression of a protein at level of cellular elements at least 3-6 samples for each tissue are required. In the manuscript the authors refer to one single subject and there is not a statistical analysis. The authors should add more additional data on this, other that they have to add more detailed information on the cell lysate preparation from fluid (saliva) and on the specificity of the antibody. Moreover, in the chapter (HCABS1 in Serum) the authors should explain better the specific selection of the quantity of antibody and serum used for analysis. Moreover, I noticed some mistake in the number of the Figure 2 and its description in the text respect to the illustration. In addition, a part of western blot images of the Figure 2 is the same just reported in the thesis (title “The perplexity of calcium-binding protein, spermatid-associated 1 (CABS1): by Eduardo Alejandro Reyes Serratos, 2022) but the authors do not report this reference. The title of the two lanes reported in this figure is not clear and the authors have to modify it.

In addition, the authors should better specify the time of the execution of analysis on saliva samples considering that the samples were collected between 2017 and 2018 and indicate n. of samples tested. Regarding the immunohistochemical studies, the authors should indicate how many slides for samples were prepared for the analysis and how many observers (considering the intra- and inter-observer variability) were used. The authors have also to reorganize better the discussion that is sometimes confuse. The authors sometimes describe some aspects that are obvious (i.e. the role of Leydig cells in the production of androgens) and not necessitate to be described.

In addition, albeit the English form requires minor revision, a particular attention is due to the discussion where some sentences are not completely clear.

6. PLOS authors have the option to publish the peer review history of their article (what does this mean?). If published, this will include your full peer review and any attached files.

Reviewer #1: No

Reviewer #2: No

---

## [Author Response · Author response to Decision Letter 0]

8 Mar 2024

Response to Reviewers:

Point-by-Point Responses

Journal Requirements:

RESPONSE: We have made extensive changes in accordance with the style requirements. 

MMP, RGPIN-2020-04553, Natural Sciences and Engineering Research Council of

Canada, https://www.nserc-crsng.gc.ca/.

ADB, 16BB-MSI-C5 and 18BB-SI-C9. AllerGen. https://www.allergen.ca/.

RESPONSE: The funders, NSERC and AllerGen, had no role in study design, data collection and analysis, decision to publish, or preparation of the manuscript.

GB Diagnostics, 10515302 Canada Inc. provided in-kind support within the company for characterization of the polyclonal and monoclonal antibodies to human CABS1 and for their long term storage. Drs Buck and Gonshor, co-owners of GB Diagnostics contributed to design of the monoclonal antibodies and analysis of their specificities, as well as to discussions about the preparation of the manuscript and its editing. 

Stephen Lewis and Michiko Watanabe have no funding to declare.

I have read the journal's policy and the authors of this manuscript have the following competing interests:

* A. Dean Befus, together with the University of Alberta holds a patent on CABS1 as a biomarker and an associated licensing agreement with GB Diagnostics.

* Aron Gonshor and Robert Buck are co-owners of GB Diagnostics.

We note that one or more of the authors are employed by a commercial company: name of commercial company. 

RESPONSE: GB Diagnostics, 10515302 Canada Inc. provided in-kind support for characterization of the polyclonal and monoclonal antibodies to human CABS1 and for their long term storage. Drs Buck and Gonshor, co-owners of GB Diagnostics contributed to design of the monoclonal antibodies and analysis of their specificities, as well as to discussions about the preparation of the manuscript and its editing.

“The funder provided support in the form of salaries for authors, but did not have any additional role in the study design, data collection and analysis, decision to publish, or preparation of the manuscript. The specific roles of these authors are articulated in the ‘author contributions’ section.”

RESPONSE: Roles of Drs. Buck and Gonshor are defined in the statement of Author Contributions and Funding Statement.

RESPONSE: AG and RB are co-owners of GB Diagnostics 10515302 Canada Inc. GB Diagnostics helped with the processing of the patent application and partially funded the approved USA patent 16/084,617, entitled: Calcium Binding Protein, Spermatid Specific 1, as a Biomarker for Diagnosis or Treatment of Stress. The company is also assisting with the application and partially funding of the Canadian National Application No. CA 3,017,604, Calcium Binding Protein, Spermatid Specific 1, as a Biomarker for Diagnosis or Treatment of Stress (pending approval).

GB Diagnostics 10515302 Canada Inc. is developing a stress biomarker test for commercial application under its licensing agreement with the University of Alberta. 

The commercial affiliation with GB Diagnostics Canada Inc. does not alter our adherence to PLOS ONE policies on sharing data and materials for this manuscript.

RESPONSE: Done 

RESPONSE: We have now provided three supplementary figure files and two supplementary tables in the manuscript, as well as a link to a data repository for the Western Blot raw images and protocol, and mass spectroscopy data (see lines 672-678). https://sites.ualberta.ca/~marcelo/Data_Repository.zip

5. Please include a caption for figure 4.

RESPONSE: We have included the caption to previous figure 4 and corrected the associated errors in the following captions. Previous figure 6 has been moved to the supplementary S2 Fig. The figures and captions are now in alignment (see comments in response to reviewer #1 also).

RESPONSE: We have made these changes as requested.

7. PLOS ONE now requires that authors provide the original uncropped and unadjusted images underlying all blot or gel results reported in a submission’s figures or Supporting Information files. This policy and the journal’s other requirements for blot/gel reporting and figure preparation are described in detail at https://journals.plos.org/plosone/s/figures#loc-blot-and-gel-reporting-requirements and https://journals.plos.org/plosone/s/figures#loc-preparing-figures-from-image-files. When you submit your revised manuscript, please ensure that your figures adhere fully to these guidelines and provide the original underlying images for all blot or gel data reported in your submission. See the following link for instructions on providing the original image data: https://journals.plos.org/plosone/s/figures#loc-original-images-for-blots-and-gels. 

RESPONSE: We have provided the original Western Blot images in the public data repository link; https://sites.ualberta.ca/~marcelo/Data_Repository.zip

RESPONSE: Done 

Additional Editor Comments:

Dear Authors,

The concept of the present study seems interesting. In this work, the authors address an interesting and original topic, which is the characterization CABS1 by using different approaches (immunohistochemistry, WB and bioinformatics) and it moves from the development of a technique just validated to obtain monoclonal and polyclonal antibodies. However, the manuscript in its actual form does not meet the standards set forth by Plos One. I highly recommend the authors to address precisely the reviewers’ concerns before delivering any decision.

Thank you,

Abeer El Wakil.

Reviewers' comments:

Reviewer's Responses to Questions

Comments to the Author

1. Is the manuscript technically sound, and do the data support the conclusions?

Reviewer #1: Yes

Reviewer #2: Partly

RESPONSE: See responses below to reviewers.

2. Has the statistical analysis been performed appropriately and rigorously?

Reviewer #1: Yes

Reviewer #2: No

RESPONSE: The manuscript does not contain any statistical analysis. 

3. Have the authors made all data underlying the findings in their manuscript fully available?

Reviewer #1: Yes

Reviewer #2: Yes

4. Is the manuscript presented in an intelligible fashion and written in standard English?

Reviewer #1: Yes

Reviewer #2: Yes

5. Review Comments to the Author

Reviewer #1: Comments about the manuscript:

“Identification and Characterization of Calcium Binding Protein, Spermatid Associated 1 (CABS1) in Selected Human Tissues and Fluids”

Calcium-binding protein associated with spermatid 1 (CABS1) is widely studied in spermatogenesis. Its mRNA is present in many other tissues, but with little information about it. The aim of the study presented here was to investigate the presence of this protein in different tissues (salivary glands, saliva, serum and testes) using polyclonal and monoclonal antibodies directed against different sections of the protein, prepared during from previous work, by Western blot, immunohistochemistry and bioinformatics.

This substantial work brings interesting and useful results. The manuscript nevertheless needs to be improved before considering its publication.

Page 5 “WB”: western blot? Write in full the first time the name appears in the text.

This remark is applicable throughout the manuscript. Please check this point. On the other hand, when the number of abbreviations is large (like here), it would be useful to provide a glossary of abbreviations.

RESPONSE: Thank you for this comment. “WB” was originally defined on page 4 in the manuscript, but in response to other comments, below, the first use of “WB” has been moved to line 117 on page 5. 

Moreover, in accordance with the request from the reviewer, we have provided a glossary of abbreviations S1 Table. This is a valuable addition to the manuscript. 

Page 6. “Post-homogenization, samples were centrifuged”: specify the characteristics of the centrifugation: Speed (in number of g), duration.

RESPONSE: This has been added as requested, line 144, page 7. 

Page 9. Immunohistochemical analyses. In the description of the technique, specify how the negative controls were carried out.

RESPONSE: Thank you for pointing out this omission. The negative controls for the mAb studies have now been included in the Immunohistochemistry section of the Materials and Methods (lines 223-225, page 10). 

Page 9. “SMG“: write in full SMG (submandibular gland, I think?) with abbreviation between brackets, when it is encountered for the first time in the text. See note above.

RESPONSE: This abbreviation was defined on page 6 of the original manuscript, and after restructuring for the revisions, it remains on page 6, line 133.

Page 12, Figure 1 - Western blot analyses of human CABS1 in transient overexpression lysate (OEL) of HEK 293T cells and in control lysate (NCL) of HEK 293T cells transfected with empty plasmid: Is this really figure 1? it looks to me like figure 2. Please check and correct.

RESPONSE: Thank you for identification of our error. It has been corrected. 

Page 12, Figure 2 – Estimate of the abundance of hrCABS1 in sections (range of kDa) of 1 D gel analyzed by mass spectroscopy: same: Is it not figure 3? Please, check and correct.

RESPONSE: Corrected, thank you.

Page 13, Figure 3 – Western blot analyses of CABS1 in human female and male submandibular gland lysates. A: Is it not figure 4? Please check and correct.

RESPONSE: Corrected, thank you. 

Between Figures 3 and 5 (Figure 5 – Western blot analyzes of CABS1 in human (male) saliva), there is no Figure 4. Please check.

RESPONSE: Corrected, thank you. 

Page 15, at the bottom of the page, “Fi

---

## [Decision Letter · Decision Letter 1]

24 Mar 2024

Identification and Characterization of Calcium Binding Protein, Spermatid Associated 1 (CABS1) in Selected Human Tissues and Fluids

PONE-D-23-29960R1

Dear Dr. Marcet-Palacios, 

We’re pleased to inform you that your manuscript has been judged scientifically suitable for publication and will be formally accepted for publication once it meets all outstanding technical requirements.

Kind regards,

Abeer El Wakil, PhD

Academic Editor

PLOS ONE

Additional Editor Comments (optional):

Reviewers' comments:

Reviewer's Responses to Questions

**Comments to the Author**

1. If the authors have adequately addressed your comments raised in a previous round of review and you feel that this manuscript is now acceptable for publication, you may indicate that here to bypass the “Comments to the Author” section, enter your conflict of interest statement in the “Confidential to Editor” section, and submit your "Accept" recommendation.

Reviewer #1: All comments have been addressed

Reviewer #3: All comments have been addressed

2. Is the manuscript technically sound, and do the data support the conclusions?

Reviewer #1: Yes

Reviewer #3: Yes

3. Has the statistical analysis been performed appropriately and rigorously? 

Reviewer #1: Yes

Reviewer #3: N/A

4. Have the authors made all data underlying the findings in their manuscript fully available?

Reviewer #1: Yes

Reviewer #3: Yes

5. Is the manuscript presented in an intelligible fashion and written in standard English?

Reviewer #1: Yes

Reviewer #3: Yes

6. Review Comments to the Author

Reviewer #1: Comments about the revised version of the mansucript:

“Expression pattern of germ cell markers in cryptorchid stallion testes”

The authors took into account the remarks I made following the first version. I thank them for that. For me, the article is ready to be published.

Reviewer #3: (No Response)

7. PLOS authors have the option to publish the peer review history of their article (what does this mean?). If published, this will include your full peer review and any attached files.

Reviewer #1: No

Reviewer #3: No

---

## [Editor Report · Acceptance letter]

25 Apr 2024

PONE-D-23-29960R1 

PLOS ONE

Dear Dr. Marcet-Palacios, 

I'm pleased to inform you that your manuscript has been deemed suitable for publication in PLOS ONE. Congratulations! Your manuscript is now being handed over to our production team.

Kind regards, 

on behalf of

Professor Abeer El Wakil 

Academic Editor

PLOS ONE